# The Complete Mitochondrial Genome of the Deep-Dwelling Goby *Suruga fundicola* (Teleostei, Gobiidae) Reveals Evidence of Recombination in the Control Region

**DOI:** 10.3390/ijms26178317

**Published:** 2025-08-27

**Authors:** Changting An, Ang Li, Huan Wang, Shuai Che, Richard van der Laan, Shufang Liu, Zhimeng Zhuang

**Affiliations:** 1State Key Laboratory of Mariculture Biobreeding and Sustainable Goods, Yellow Sea Fisheries Research Institute, Chinese Academy of Fishery Sciences, Qingdao 266071, China; anct@ysfri.ac.cn (C.A.); liang@ysfri.ac.cn (A.L.); wanghuan@ysfri.ac.cn (H.W.); cheshuai@ysfri.ac.cn (S.C.); zhuangzm@ysfri.ac.cn (Z.Z.); 2Laboratory for Marine Fisheries Science and Food Production Processes, Qingdao Marine Science and Technology Center, Qingdao 266237, China; 3Independent Researcher, Grasmeent 80, 1357JJ Almere, The Netherlands; richard.vanderlaan80@gmail.com

**Keywords:** *Suruga fundicola*, mitogenome, long tandem repeats, mitogenome recombination

## Abstract

*Suruga fundicola*, one of the few known deep-dwelling gobies, is found in Japan, South Korea, and China. Owing to the limited availability of specimens, little is known about its mitogenome characterization, phylogenetic relationship, and adaptive evolution. In this study, we sequenced four complete mitogenomes using the DNBSEQ platform and Sanger sequencing. The mitogenomes in length ranged from 17,138 to 17,352 bp, primarily due to the variation in the number of long tandem repeat (LTR) sequences within variable region 3 (VR3). Although the gene composition and arrangement of the *S. fundicola* mitogenome are largely consistent with those of other gobies, we identified an expansion of the *ND2* gene (78 bp), and an unexpected noncoding region (NC, 35 bp) located between the *ND2* and *tRNA^trp^* genes. To further investigate the variation in VR3, we sequenced this region in all nineteen individuals with the Sanger sequencing method. We detected eight distinct LTR types, containing one–three mutation sites, which formed ten different VR3 patterns. Most VR3 patterns (14/19) consisted of a single type of pure LTR, while the remaining five exhibited heterogeneous patterns composed of two different LTRs. Notably, in LTR types T1 and T3, which co-occur in heterogeneous patterns P1 and P9, we found their respective pure patterns (P2–3 and P7). Recombination provides a better, more plausible mechanism for generating the heterogeneity patterns than slipped-strand mispairing, which better explains the homogeneous LTR expansions. These findings provide evidence of recombination in the control region of a vertebrate mitogenome. A phylogenetic analysis confirmed that *S. fundicola* has a close relationship with *Am. hexanema* and *C. stigmatias*. Compared to five shallow-water species of the AcanthogobiusGroup, the deep-dwelling goby *S. fundicola* was found to be under stronger purifying selection. Within its mitochondrial protein-coding genes (PCGs), *ND2* and *ND6* genes were subject to stronger purifying selection than the others. Additionally, four genes showed signs of selection sites with high credibility (one in *ATP6*, *ND3*, and *ND4*; eight in *ND2*). This study provides valuable genomic resources for *S. fundicola* and enhances our understanding of the phylogenetic relationship, mitogenome recombination, and adaptive evolution of the goby.

## 1. Introduction

Mitochondria are essential cellular organelles for life found in almost all eukaryotes, and they contain their own genetic information, the mitogenome [1]. It has generally been assumed that this mitogenome does not undergo recombination, but this assumption has been challenged by a few cases of heteroplasmy and mtDNA recombination, as already reported for mussels [2,3], fishes [4,5], birds [6], cats [7], and humans [8,9]. Genes typically encoded in the vertebrate mitogenome contain 13 proteins, 2 ribosomal RNAs (rRNAs), and 22 transfer RNAs (tRNAs) [10,11,12]. Two noncoding regions (NC), the control region (CR), and the origin of L-strand replication (OL), were identified in almost all vertebrate mitogenomes [13]. The CR contains regulatory elements for controlling the transcription and replication of the mtDNA molecule, such as the termination-associated site (TAS), conserved sequence blocks (CSBs I to III), and variable regions (VR1, VR2, and VR3); see Figure 1 [13]. The length of the CR is highly variable due to the appearance or absence of variable numbers of tandem repeats (VNTRs). Tandem repeats have been widely reported in the mitochondrial CR of animal species, and variations in the number of repeat units have been found in birds [6], owls [14], cats [7], rabbits [15], and fishes [4,16,17]. Moreover, an extreme variation in the number of repeats has been found in interspecific [16], intraspecific [7,15], and intra-individual levels [14,18].

Based principally on surveys of the axial skeletal features of gobiid fishes, *Suruga fundicola* Jordan & Snyder, 1901, was placed into a putative monophyletic group (the Acanthogobius Group) with seven other genera [19,20]. However, due to the characteristics of gobiid fishes—exceptional species diversity (more than 2400 species), broad biogeographic distribution, diminutive body sizes, and heterogeneous morphological specialization/degeneration—significant challenges arise in their traditional morphological classification [21,22,23]. With the help of molecular evidence, the monophyly of the Acanthogobius Group was verified by recent studies (although based on a few species of the group), and the group was confirmed to be part of a larger clade, the *Acanthogobius* lineage (a monophyletic lineage within the family Gobiidae). This clade comprises about 29 genera, mainly living in the Northern Pacific [21,22,23]. McCraney, Thacker, and Alfaro further confirmed the monophyly of this group based on more taxa [23]. The monophyly of the Acanthogobius Group was recently challenged, although this study was based on only a few mitogenomes [24]. In our prior study, a preliminary phylogenetic analysis of concatenated *COX1* and 12S sequences (704 bp) incorporating *S. fundicola* failed to recover a monophyletic Acanthogobius Group. This likely resulted from an insufficient number of phylogenetically informative sites due to the limited sequence length [25]. Thus, it is a better choice to use more mitogenomes to investigate both the phylogenetic status of *S. fundicola* and the monophyly of the Acanthogobius Group.

*Suruga fundicola* inhabits temperate regions of the Northwest Pacific, including Japan, South Korea, and China. This species has been occasionally recorded in marine surveys as one of the deepest-dwelling gobies [25,26,27,28]. Compared to other shallow-water species of the Acanthogobius Group, *S. fundicola* occupies significantly deeper habitats (depths of about 40–400 m) [20,28], enduring high hydrostatic pressure, low temperatures (mean: 8 °C), and permanent low-light conditions [25,29]. Down to the limits of daylight (1000 m) in the sea, the eyes of fishes generally increase in size relative to body length [29]. The eyes of *S. fundicola* are notably larger than those of other fishes in this group [19,28], clearly an adaptation to this deepwater environment. The harsh deepwater conditions are expected to result not only in the morphological traits but also in the adaptive evolution of the protein-coding genes that are vital for the oxidative phosphorylation pathway [7]. Intriguingly, energy-intensive species like bats exhibit much higher selection pressures on mitochondrial-encoded OXPHOS genes relative to their nuclear counterparts [30]. The positive selection sites identified in *Triplophysa* cavefishes indicated adaptive evolution in mitochondrial PCGs in response to extreme subterranean conditions [31]. However, whether mitochondrial proteins in this deep-water goby have undergone adaptive evolution remains unknown.

To date, only five mitogenomes (a total of ~150) of gobies within the Acanthogobius Group have been deposited in the MitoFish database (https://mitofish.aori.u-tokyo.ac.jp/species/search/ accessed on 20 May 2024). Few published studies have mainly involved the mitogenome architecture and phylogenetic relationships of only a few species [24,32,33], and have also addressed the interspecific variations between *Chaeturichthys stigmatias* and *Amblychaeturichthys hexanema* [24]. Due to the absence of mitogenomic data for the deep-dwelling goby, its mitochondrial architecture, phylogenetic relationships, and deepwater adaptive evolution remain poorly characterized. Thus, we sequenced four complete mitogenomes of *S. fundicola* and the VR3 part of the mitogenomes of nineteen individuals, based on next-generation sequencing (NGS) and/or the Sanger sequencing method. Firstly, we characterized the basic mitogenomic structure of this species and the arrangement of the various mt-genes. Secondly, we performed a comparative analysis with other goby fishes, and we analyzed the phylogenetic relationship of the group within the Gobiiformes. Finally, we analyzed the evolutionary selection pressure on 13 proteins coded by the mitogenome of *S. fundicola*.

## 2. Results and Discussion

### 2.1. General Features of S. fundicola Mitogenome

The *ND2* gene and tandem repeats of the four mitogenomes were once more determined by the Sanger sequencing method. The four complete mitogenomes of *S. fundicola* and four sequences of the *ND2* expansion were submitted to GenBank with the accession numbers PV067722-25 and PV067729-32. The four circular mitogenomes ranged from 17,138 to 17,352 bp in length, the sizes of which were longer than most mitogenomes of known goby fishes. All the typical 37 genes, including 13 PCGs (*ND1–6, ND4L, COX1–3, CYTB, ATP6,* and *ATP8*), 22 tRNA genes, and 2 rRNA genes (*12S rRNA* and *16S rRNA*) were identified. Most of the genes were encoded on the H-strand; only *ND6* and 8 tRNAs were on the L-strand (Figure 2, Table 1). The identified 37 genes of the four mitogenomes had all the same length and relative position. An unexpected NC (36 bp length in PV067722 and 35 bp length in PV067723-25) was identified between *ND2* and *tRN^Trp^*, which has not been found in other fishes (as far as we know).

The base composition of the complete mitogenome (PV067722) was as follows: A% = 28.6, T% = 27.6, C% = 27.0, G% = 16.8 (Table 2). The overall A + T content was 56.4%, which was slightly richer than the G + C content (43.6%). The overall GC- and AT-skews of the mitogenome were −0.23 and 0.02, respectively; thus, indicating a strand compositional bias characterized by a substantial excess of over G nucleotides and a slight excess of over T nucleotides, similar to other fishes [34,35].

The PCGs accounted for 66.7% (11, 506 bp) of the *S. fundicola* mitogenome, with nucleotide composition of the PCGs: T% (30.4%), C% (28.5%), A% (24.8%), and G% (16.3%), with an AT content of 55.2%. The start codon ATG was found in 12 of the 13 PCGs, only *COX1* was initiated with the codon GTG. The stop codons of the 13 PCGs are mostly TAA (*ND2*, *COX1*, *ATP8*, *ND4L, ND5*) and include TAG (*ND1*, *ND6*), TA (*ATP8*), and T (*COX2*, *COX3*, *ND3*, *ND4*, *CYTB*).

The 13 protein-coding genes have 3839 codons. Leu, Ala, Thr, and Ile were the most abundant amino acids in the mitogenome PCGs (Figure 3). Codon frequencies were calculated by the occurrence frequency of codons in all PCGs. Leu (CUU, CUA), Ala (GCC), Arg (CGA), Gly (GGC), Phe (UUU), and Thr (ACC) were the most abundant codons; meanwhile, stop codons UAG and UGU were the least frequent. The relative synonymous codon usage (RSCU) is a widely used measurement for the codon bias of each amino acid. The RSCU value for a codon close to 1 means that there is no preference for that codon, and an RSCU value >1 indicates that there is a strong bias for that codon. The RSCU values of the *S. fundicola* genome are displayed in Figure 4. Pro (CCC), Arg (CGA), Ala (GCC), Thr (ACC), Ser (UCC), and Leu (CUC and CUA) were the most frequently used codons, while Ser (UCG and AGU), Pro (CCG), Ala (GCG), Thr (ACG), and Leu (UUG) were the least.

Two common noncoding regions, the OL and the CR, were similar to those reported in other fish species [13]. The OL was 33 bp in length, located in the WANCY cluster between *tRNA^Asn^* and *tRNA^Cys^*. The CR was 1495 bp in length, only slightly less than the CR of *Am. hexanema*, but longer than any other gobies of the Acanthogobius Group. Three termination-associated sequences (TAS), two conserved sequence blocks (CSB II to III), and one T-homopolymer were identified, but we could not identify the CSB-D block [13,16]. The LTR was 107 bp in length, consisting of an upstream part (52 bp) and a downstream part (55 bp), as shown in Figure 5. The homolog of the upstream part can be found in the CR of other species, and the homolog of the downstream part is its *tRNA ^Phe^*. In addition, a 5′ imperfect copy was found in all individuals downstream of the CR, which includes the complete upstream part and the first five bases of the downstream part. In addition, the LTR can form a second structure with a higher free energy dG = −11.8 (Appendix A), and complex structures with multiple stem-loops were found between two LTRs with a higher free energy dG = −21.57 (Appendix A). The complex structures can help to form the VRNT [6].

### 2.2. Variation in the Long Terminal Repeat

The nineteen LTR sequences in the VR3 with lengths ranging from 582 to 930 bp were submitted to GenBank with the accession numbers PV077033-51 (Appendix A). The number of LTRs in the sequenced nineteen individuals was very variable: three (once), four (12 times), 5 (5 times), and six (once), see Figure 6b. In the phylogenetic tree, the individuals with the same LTRs were distributed in different clades, the changes in the LTRs did happen much faster than changes in the nuclear base composition. At the same time, the individuals from the same location were clustered in different clades. This phenomenon is also observed with yellow-browed tits [6] and domestic cats [7], but these studies differ from ours: these variable repeats are related to pedigree and geographic groups [36,37,38,39]. In our study, the observed variation in the LTRs was not a suitable marker for discussing lineage divergence. The LTRs have been observed in several vertebrates, and extreme variation in the number of repeats has been found in many intraspecific [6,7] and intra-individual studies [14,18,40]. The structure and functions of the coding regions are well studied, but the CR’s function is not entirely clear. It has been demonstrated that the CR region holds the promoters for the replication; however, the function of a series of tandem repeats is still unknown [15].

There were eight types (T1–T8) of LTRs found in the VR3 (Figure 6a), six with a single variation point compared to T1, one (T6) with 2, and one (T4) with 3 points. The eight types formed ten different patterns (P1–P10), five of which have only one type of LTR, and the other five have two heteroplasmic types of LTRs. The T1 LTR can be found in six different patterns (P1–3, P5–6, and P9), and the T3 LTR can be found in three different patterns (P1, P7, and P9). Most duplication events in mtDNAs can be explained by slipped-strand mispairing [6,41,42]. Considering that two LTRs can form complex structures, slipped-strand mispairing would be the best explanation for the patterns with one type of LTR. Slipped-strand mispairing was not a good explanation for the occurrence of the heteroplasmic patterns, especially considering that the LTRs occupied by P1, P5, P6, and P9 can be identified in other patterns (individuals) as well. That is to say, the “P2” and “P7” type mitochondria must be present in the same cell, and their DNA must be coupled. This implies paternal leakage followed by fusion of the mitochondria [3]. Insertion via recombination is probably a better explanation for the origin of these heteroplasmic LTRs. This heteroplasmy in the LTRs at the intraspecific level was also found in a variety of species [3,4,5,6,7,8,9], and direct evidence for recombination has already been discovered in two fishes, the Atlantic Cod (*Gadus morhua*) [4] and the European Flounder (*Platichthys flesus*) [5]. Here, we present another case for recombination in the mtDNA of a vertebrate.

### 2.3. Comparative Analysis

#### 2.3.1. A + T% and AT-Skew of the Acanthogobius Group Mitogenomes PCGs

The A + T content and AT-skew of the 13 mitogenome PCGs of the Acanthogobius Group were calculated (six gobioid species; see Figure 7a). The six species all exhibit AT bias. The A + T content of *S. fundicola* (55.21%) was significantly higher than that of the other species (*p* < 0.01), with a 95% CI of [53.43, 55.08]. The value of *A. flavimanus* was the lowest. The AT-skew of *S. fundicola* was significantly lower than that of the other five species (*p* < 0.01), with a 95% CI of [−0.11, −0.04], see Figure 7b.

#### 2.3.2. Transfer RNA of the Acanthogobius Group

The typical 22 tRNAs were identified in the Acanthogobius Group mitogenomes, as in other fishes. Taking *S. fundicola* as an example, the secondary structure of tRNAs generally contains four domains and a short variable loop: the AA (amino acid acceptor) stem, the dihydrouridine arm (D stem and loop, D), the thymidine arm (T stem and loop, T), the anticodon arm (AC stem and loop, AC), and the variable loop (V; Figure 8). Among the tRNAs, *tRNA^S^^er1 (AGC)^* is the only one that is not folded into the typical clover-leaf secondary structure, due to the absence of the DHU arm, which is similar to the structure found in other fish [13,43]. In addition, *tRNA^Cys^* is supposed to have lost the D-loop [44]. Unmatched base pairs in the stem are widespread among tRNAs; only three (*tRNA^Ala^*, *tRNA^Gln^*, *and tRNA^Tyr^*) have a fully paired stem in the investigated mitogenomes. Compared to the standard *tRNA^Phe^*, the putative “*tRNA^P^^he^*” identified in the lower region of the CR does not have the AA stem, and the code “GAA” is changed into “AAA”.

#### 2.3.3. ND2 Gene of the Acanthogobius Group Mitogenomes

Compared to the other five species of the Acanthogobius Group, the length of the *ND2* gene of the *S. fundicola* is 1122 bp, obviously longer than the length of 1044 bp of the other five members of the Acanthogobius Group. The *ND2* gene of *S. fundicola* has a unique expanded 78 bp sequence at the end of this gene, the homologous sequence of which was not found by the BLASTn tool in GenBank.

Most motifs of the *ND2* gene are similar, except for Nos. 20, 22, 24, and 25 (Figure 9). The No. 24 motif cannot be identified in *S. fundicola*, *Am. hexanema*, and *C. stigmatias*. The No. 20 motif in *Lophiogobius ocellicauda* was transferred to the upper side of the No. 17 motif. The No. 22 motif was not found in *S. fundicola*, and No. 25 was found in the expanded region of the *ND2* gene of *S. fundicola*.

#### 2.3.4. Gene Arrangement Analysis of Goby Fish Mitogenomes

The typical 37 genes (13 PCGs, 22 tRNAs, and 2 rRNA genes) and the two noncoding regions (CR and OL) were identified in 150 gobies’ mitogenomes, only the OL of *E. klunzingerii*, was not identified (Figure 10 type VII). The 150 goby fishes mitogenomes display nine types of gene arrangement (Figure 10). Most species (*n* = 138, 92.0%) of the goby fishes compared in the present study had the typical gene order, widely shared among vertebrate mitogenomes. However, deviations from the conserved gene order were found in twelve species from four families (12/150 species = 8.0%). All of the gene rearrangements mainly involve local position changes (reshuffling) or displacement to another location (translocation). Switching of the encoded strand (inversion) was not found in this study [13].

The local reshuffling was identified within three tRNA gene clusters (IQM, WANCY, and HS2L2). The tRNA gene order IQM was mutated to QIM in types V, VI, and IX. The relative location of gene WANCY was changed to NC and OL in types II and IX, respectively. The gene order HS2L2 in type VIII was changed into S2L2H. The translocation was found in type IX. The tRNA gene F was duplicated in types II, IV, VI, and IX.

#### 2.3.5. Start and Stop Codons of the Mitogenomes of Gobies

The 13 protein-coding genes of 150 goby fishes were found to possess eight types of start codons (Figure 11; Appendix A). Among these eight codons, ATG was predominant (90.7%) as already found in other fishes [13], and it was found exclusively in the *COX2*, *COX3*, *ND3*, and *CYTB* genes. GTG was only found in the *COX1* gene as a start codon. Of the eight codons, CTA was identified as a start codon in the *ND1* gene of the *Rhinogobius leavelli* (MH729000, Oxudercidae), which was the first example reported in a goby [13,43]. The code ATT was only found in the *ND1* gene of *Brachygobius doriae* (KU674800, Oxudercidae), and ATC was only found in the *ND5* gene of *Tridentiger trigonocephalus* (KT282115, Oxudercidae).

The stop codons TAA (41.3%) and T-- (39.2%) were most common in gobies, a result usually encountered in other fishes [13,41]. AG- was found once in the *ND4* gene by *Odontobutis haifengensis* (MF383619, Odontobutidae). The TA-, T-, and AG- stop codons were found mostly when the 3′-end of the protein-coding genes (i.e., *COX2*, *ATP6*, *COX3*, *ND3*, *ND4*, and *CYTB*) was followed by a tRNA gene on the same strand. The complete stop codons were found in the *ND5*, *ND6*, and *COX1* genes. The most common stop codon of *ND2* is TAA (56.0%), which differs from other fishes (91.2%) [13]. Only one stop codon, TAA, was found in *ND4L*, and seven stop codons (T--, TA-, TAA, AGA, AG-, AGG, and TAG) were found in the other genes (Figure 11, Appendix A).

### 2.4. Phylogenetic Inferences of Gobies

Molecular phylogenetic analyses were performed using 13 PCGs from 150 mitogenomes of Gobiiformes. The ML and IQ analyses generated similar topologies (Appendix A and Figure 12). In the ML tree, the species of gobies were divided into five clades (Gobiidae, Eleotridae, Odontobutidae, Rhyacichthyidae, and Trichonotidae), which was similar to recent studies [10,23]. The species of Eleotridae did not cluster into a monophyletic group but were divided into two clades: Eleotrinae (I) and Butinae (II). The latter clade was sister to the clade Gobiidae [10,45]. The species of Gobiidae formed a monophyletic group and clustered into two main clades: Gobiine-like (III) and Gobionelline-like (IV).

The Acanthogobius lineage is closely related to a clade consisting of the species of *Gymnogobius*, *Luciogobius*, *Chaenogobius*, *Eucyclogobius,* and *Gillichthys*, a result similar to recent studies [16,23,45]. Relationships within the Acanthogobius lineage recovered by our analysis conflicted with results from molecular phylogenies incorporating nuclear genes [21,22,23] or single-copy nuclear coding markers [45] but were consistent with the study based on mitogenome PCG sequence [10]. The species of the Acanthogobius Group formed a monophyletic group, containing *A. hasta*, *A. flavimanus*, *L. ocellicauda*, *S. fundicola*, *Am. Hexanema*, and *C. stigmatias*, as found in recent studies [16,21,23,45], with the exception of two other studies [24,27]. *S. fundicola* was sister to the clade formed by *Am. hexanema* and *C. stigmatias* (BP 100%), a result already found by An et al. [25].

### 2.5. Selective Pressure Analysis

The Ka/Ks ratio values were calculated to represent the selective pressure magnitude and direction of the PCGs. The average Ka/Ks values of 13 PCGs of the Acanthogobius Group were <1, suggesting a purifying selection of functional genes. Among the six species, *S. fundicola* has a lower value than other species of this group (Figure 13), indicating a stronger purifying selection, resulting in the accumulation of less nonsynonymous mutations [46,47]. Our results of the Ka/Ks comparison between *S. fundicola* and the other Acanthogobius Group species showed that *ATP8* (Ka/Ks < 0.35; *p* < 0.05), *ND2* (Ka/Ks < 0.20; *p* < 0.05), and *ND6* (Ka/Ks < 0.20; *p* < 0.05) were the genes with the highest Ka/Ks values (Figure 14). This result indicates less intense purifying selection compared to the remaining mitochondrial PCGs [48]. In turn, *COX1* possesses the lowest Ka/Ks values, indicating the strongest purifying selection, resulting in the accumulation of less nonsynonymous mutations. According to the literature, purifying selection is the predominant force acting during the molecular evolution of mitogenomes [47,48,49].

The result of signatures of positive selection in all 13 mitochondrial PCGs based on BM in EasyCodeML is presented in Table 3. All mitochondrial PCGs have values of ω < 1, with Model 0. Two genes, *ND2* (ω = 0.182; *p* = 0.0489) and *ND4* (ω = 0.071; *p* = 0.026), provided a statistically significant better fit than the one-ratio model. According to this result, the two genes (*ND2* and *ND6*) experienced stronger purifying selection than the other mitochondrial PCGs.

The BSM analysis in EasyCodeML v.1.41 was used to detect signatures of positive selection in *S. fundicola* (as foreground branch) compared to the other five Acanthogobius Group species (background branches). Four genes were found to have positive selection sites (one in *ATP6*, *ND3*, and *ND4*; eight in *ND2*), based on the MA, although this result did not receive the support of the *p*-value of the LRTs (>0.05) (Table 3). These sites were mapped onto the 3D structures of corresponding homologous proteins (marked with red color in Appendix A). Most sites were located within the functional domains of α-helices, particularly near or on the junction sites of the α-helices and loop areas, which were probably crucial for the conformational stability of the relevant proteins. The mitochondrial respiratory chain generates 95% of the adenosine triphosphate (ATP) required for cellular activities. All 13 mitochondrially encoded proteins participate irreplaceably in electron transport and aerobic respiration [50]. Studies have demonstrated that mitochondrial OXPHOS genes in bats undergo significantly stronger selective pressure than nuclear-encoded counterparts (23.08% vs. 4.9%) [30]. In this study, signals of positive selection were detected in four mitochondrial OXPHOS genes. *ATP6* encodes subunit a of ATP synthase Fo, directly driving ATP synthesis, and *ND2–ND4* constitute core functional units of the NADH dehydrogenase complex (Complex I), pivotal for energy transduction. Compared to shallow-water gobies of this Acanthogobius Group, *S. fundicola* and cavefishes (e.g., *Triplophysa*) face analogous extreme environmental pressures (high hydrostatic pressure, darkness, extreme temperature, and food scarcity) [25,31]. Notably, all four positively selected genes identified here also exhibit adaptive signatures in cavefishes. *ND2*—the gene with the highest number of positively selected sites in *S. fundicola* (eight sites)—also shows multi-site adaptive evolution in *Triplophysa* (six sites). We thus propose that in *S. fundicola*, the *ND2* gene likely experiences dual selective pressures (purifying selection coupled with positive selection) to adapt to extreme habitats. Furthermore, accumulating evidence also reveals positive selection in mitochondrial genes under extreme environments, including deep-sea fishes [51], deep-sea mussels [52], deep-sea crab [53], and plateau mammals [54].

## 3. Materials and Methods

### 3.1. Sample Collection and DNA Extraction

During the 2021–2023 R/V cruises in the Yellow Sea, a small Agassiz trawl was employed to catch the samples, at an average speed of 3 n mile/h (=5556 m/h) for 20 min. A total of nineteen specimens of *S. fundicola* were collected from the South Yellow Sea, during the 2021–2023 R/V cruises in the Yellow Sea (Appendix A). The muscle of the right lateral dorsal was sampled and stored in anhydrous alcohol for subsequent DNA extraction. The voucher specimens attached with a unique number were fixed in 10% formalin preservative for morphological examination and also for permanent curation and were deposited in the National Marine Fishery Biological Germplasm Resource Bank, China.

Genomic DNA was extracted using TIANamp Genomic DNA Kit (Tiangen DP324, Biotech, Beijing, China), according to the manufacturer’s protocol. The quality was estimated at wavelength 260/280 nm by a Nano-300 micro-spectrophotometer (Allsheng, Hangzhou, China).

### 3.2. Library Construction and DNA Sequencing

The total mitogenome of S. fundicola was sequenced on the DNBSEQ platform by MGI Tech Co., Ltd. (Hongshan District, Wuhan, Hubei, China). To confirm the expansion of the *ND2* gene, one pair of primers [4707 (S) F: 5′-CCTGGCCAAGTCATGAACCA-3′, 5583 (S) R: 5′-CAACGTGTGATTGCCACAGG-3′] was designed for the Sanger sequencing method, with the Primer-BLAST tool (https://www.ncbi.nlm.nih.gov/tools/primer-blast/index.cgi?LINK_LOC=BlastHome, accessed on 1 May 2024). The PCR product can cover the region of the *ND2* expansion. PCR was conducted in 25 μL volumes, including 12.5 μL Master mix Taq, 1 μL of each primer, 1 μL template DNA, adding double distilled water to adjust the volume (Table 4). Thermocycling conditions were as follows: initial denaturation for 4 min at 94 °C, denaturation for 50 s at 94 °C, annealing for 50 s at 55 °C, and extension 50 s at 72 °C. After 35 cycles, the final extension was performed at 72 °C for 10 min. The PCR products were checked by the uncropped gel photograph method. The PCR products were bidirectionally sequenced by BGI Genomics Co., Ltd (West Coast New Area, Qingdao, Shandong, China).

The existence of LRT in the VR3 region can cause errors in mitochondrial genome assembly, manifested as varying VR3 lengths in repeated assembly runs using NOVOPlasty under identical parameters. The primers were designed similarly to *ND2*, the sequences are labelled as follows: 16068TR(S) F: 5′-TTCAATCGCATCTCAGAGT-3′, 75TR (S) R: 5′-CGGGAAGGAATATGTAGGG-3′. To investigate the VR3 region across different individuals, the region of all nineteen individuals was sequenced using this pair of primers. The PCR reaction conditions are detailed in Table 4 and all amplified sequences were submitted to GenBank, with the help of BankIt (https://www.ncbi.nlm.nih.gov/WebSub/, accessed on 30 January 2025).

### 3.3. Sequence Assembly, Annotation, and Analysis of S. fundicola

NOVOPlasty 4.3.5 was used for the assembly of the four mitogenomes in Linux, with the template (seed) of the *COX1* sequence [55], and the SeqMan of Lasergene. v. 7.1 (DNASTAR software package) was used to assemble the sequences generated from the Sanger sequencing method. The *ND2* and VR3 sequences of mitogenomes were confirmed in MEGA X, based on ClustalW. The mitogenomes were annotated with Mito Annotator [56]. The identification and secondary structure of tRNA were conducted in tRNAscan-SE version 1.21 [57], and the tRNA secondary structures were visualized using the Forna web server [58]. The circular mitogenome map was visualized using Proksee [59]. Base composition and amino acid distributions were calculated with MEGA X [60]. The mitochondrial genes’ skewness was calculated using the following formulas: GC-skew = (G − C)/(G + C) and AT-skew = (A − T)/(A + T).

The CR sequences were separately extracted, based on the annotation with Mito Annotator [56]. The length, A + T content, and AT-skew were calculated with MEGA X [60]. The number of repeats of CR was investigated with the web server Tandem Repeat Finder v. 4.09 (http://tandem.bu.edu/trf/trf.html, accessed on 10 October 2023) [61]. The Mfold web server (http://mfold.rna.albany.edu/?q=mfold, accessed on 15 October 2023) was employed to predict its secondary structures. Based on the aligned sequences with ClustalW in MEGA X [60], we identified the termination-associated site (TAS), conserved sequence blocks (CSB I to III), and T-homopolymer with Jalview 2.9.0b2 [13,62,63,64]. Here, we identified TAS motifs using the query ATGN (8–9) CAT/TAT and searched for CSB-like motifs based on the following queries: CSB-D: TTC/A (11)T/CTC, CSB-I: ATA (15) ATA; CSB-II: AAACCCCNNNNCCCCC; CSB-III: AAACCCCC [13,16,62]. The T-homopolymer was also recognized based on a continuous “TTTTTT” base. The Mfold web server (http://mfold.rna.albany.edu/?q=mfold, accessed on 25 October 2023) was employed to predict L-strand origin (O_L_), secondary structures [65], and the conserved teleost motif “5′-GCGGG-3′” was identified [11,13]. The holotype of the LTR in the 3′ terminal of the CR was identified in DnaSP 6 [66].

### 3.4. Comparative Analysis of the Goby Mitogenomes

For the Acanthogobius Group, we downloaded the five available goby fish mitogenomes of this group from Mitofish (http://mitofish.aori.u-tokyo.ac.jp/species/search/, accessed 20 May 2024). The base compositions of the PCGS of the Acanthogobius Group mitogenomes were calculated as in the *S. fundicola* (the PV067722 mitogenome was chosen as an example). To further examine whether these two parameters exhibit significant differences between *S. fundicola* and other species of the Acanthogobius Group, we performed one-sample t-tests using SPSS 27. The motifs of the *ND2* gene of this group were identified by MEME-SUITE 5.5 (https://meme-suite.org/meme/tools/meme, accessed on 22 August 2024), based on the maximum number of 25 motifs.

We downloaded the other 149 Gobiiformes genomes from Mitofish and annotated the genes with the annotation function in the same database. We analyzed the gene arrangement of these Gobiiformes fishes. In addition, the start and stop codons of 13 PCGS were calculated with MEGA X [60].

### 3.5. Molecular Phylogenetic Analysis

The sequences of the 3′ terminal CR of nineteen individuals were aligned and trimmed in MEGA X. These sequences of the partial CR were 571 to 898 bp in length, which consisted of one section before LTR (with a length of 256 bp) and the LTRs (with a length from 321 bp to 642 bp). The NJ method was chosen to construct the phylogenetic tree with the same software, based on the Kimura 2-parameter [60].

The taxonomic position of *S. fundicola* was assessed using concatenated sequences of 13 PCGs (without stop codon) of 150 Gobiiformes mitogenomes, which represented 74 genera from 6 families of Gobiiformes. *Kurtus gulliveri* was chosen as an out-group to the Gobioidei (Appendix A). The taxonomy of Eschmeyer’s Catalog of Fishes was followed [67]. We performed sequence alignment using the MAFFT program (alignment mode: code, code: vertebrate mitochondrial code, strategy: auto [68] and multi-gene datasets concatenated by SequenceMatrix [69]. The aligned DNA sequences were used for the phylogenetic analysis. Maximum likelihood trees (ML trees) were constructed using concatenated data (12,276 bp) based on the GTRGAMMA model in RaxML v8.2.8 [70] with 1000 bootstrap replicates. Data were partitioned by codon position and by a partitioning scheme identified in PartitionFinder v2.1.1 [71]. Initial data blocks of codons were analyzed under the models (GTR, GTR + G, GTR + G + I) and with the clustering algorithm in PartitionFinder v2.1.1 [72]. In addition, IQ-TREE multicore version 2.1.4-beta was employed to infer an ML tree for comparison, with the best-fit model GTR + F + R10 as suggested by ModelFinder [73], based on the BIC value with 1000 SH-aLRT (Shimodaira–Hasegawa-like approximate likelihood ratio test) [74]. The resulting ML and IQ trees were visualized using Figtree v.1.4.3 [75].

### 3.6. Selective Pressure Analyses

The selective pressure magnitudes and directions of 13 concatenated PCGS sequences of the six Acanthogobius Group species, the number of non-synonymous substitutions per non-synonymous site (Ka), the synonymous substitutions per synonymous site (Ks), and the Ka/Ks ratio were determined [76]. For this analysis, Ka and Ks values were calculated between each Acanthogobius Group species and the other 149 goby species, based on the 13 concatenated PCGS sequences (without stop codon). To investigate the pressures acting on each of these PCGs of the Acanthogobius Group. The Ka/Ks ratio of each PCG between *S. fundicola* and other Acanthogobius Group species was determined with the same method [76].

The EasyCodeML v.1.41 was employed to examine for signatures of positive selection in all 13 mitochondrial PCGs that could be altered by the colonization of deep-water environments in *S. fundicola* [77]. We used the phylogeny obtained with MEGA X as input files based on each gene sequence of the PCGs.

We first used Branch Models (BMs) to test for statistically significant differences in ω among branches of the phylogenetic tree. Foreground branches correspond to *S. fundicola* living in deep-water habitats. In turn, background branches are those leading to the five other species inhabiting the bottom depth less than 90 m. The one-ratio model (M0) assumes that ω is constant for all branches. The free-ratio model (M1) allows each branch to have an independent ω, and the two-ratio model (M2) assumes that the branches of interest (i.e., the foreground lineages) have different ω values than the background lineages. Here, M1 and M0 were compared to test whether different lineages in the tree had different ω values, and M2 was compared with M0 to determine whether *S. fundicola* was subjected to more selection pressure than the other Acanthogobius Group species. Pairwise models were compared using likelihood ratio tests (LRTs) to test whether M1 or M2 fit the data significantly better than M0 [78]. Considering that positive selection often occurs over a short period of evolutionary time and/or at a few sites, we chose to use branch-site models (BSM) in EasyCodeML to investigate the case of positive selection for the deep-dwelling species *S. fundicola* with the help of two models, model A and model Anull [77,79]. The three-dimensional structure of the proteins was predicted with Phyre2 [80], and the positive sites were marked with PyMOL.

## 4. Conclusions

This study sequenced and assembled four complete mitogenomes of this deepwater goby and the variable region 3 (VR3) in the 3′ terminal control region (CR) of nineteen specimens. The lengths of the four mitogenomes of *S. fundicola* ranged from 17,138 to 17,352 bp, with size variation attributed primarily to differences in the number of long tandem repeats (LTRs) within the CR. The gene composition and arrangement of the *S. fundicola* mitogenome are not significantly different from those of other gobies, but an expansion of the *ND2* gene was identified. Meanwhile, an unexpected NC was also identified between the *ND2* and *tRNA^Trp^* coding regions. Using species-specific primer/probe design targeting unique extended regions (e.g., ND2 and the NC), facilitating definitive species identification, quick species identification can be achieved without sequencing. Among gobiid mitogenomes, nine distinct gene arrangements were observed, mainly involving local position changes (reshuffling) or displacement to another location (translocation). Analysis of VR3 showed that insertion via recombination provides a better explanation for the heterogeneity patterns LTR T1 and T3, compared with the explanation of slipped-strand mispairing for pure LTR. The analysis also reveals evidence of recombination in the vertebrate mitochondrial control region. Phylogenetic reconstruction based on 13 protein-coding genes (PCGs) from 150 gobioid mitogenomes confirmed the monophyly of the Acanthogobius Group, *S. fundicola* formed a sister group relationship to the clade consisting of two other gobies, *Am. hexanema* and *C. stigmatias*. These results contribute to the systematic classification of these gobies. Selection pressure analysis indicated that *ND2* and *ND6* underwent stronger purifying selection than the other mitochondrial PCGs. Four genes harbored high-confidence positive selection sites (one site each in *ATP6*, *ND3*, and *ND4*; eight sites in *ND2*). These candidate mutations can provide prioritized targets for future mitochondrial proteomics.

## Figures and Tables

**Figure 1 ijms-26-08317-f001:**
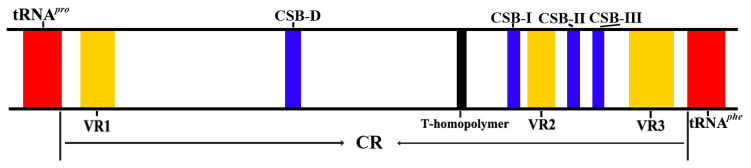
A schematic diagram of the control region (CR) of the fishes’ mitogenome, following Satoh et al. (2016) [13]. The locations of conserved sequence blocks (CSBs) and variable regions (VRs) are mapped. The T-homopolymer region is represented by a black bar.

**Figure 2 ijms-26-08317-f002:**
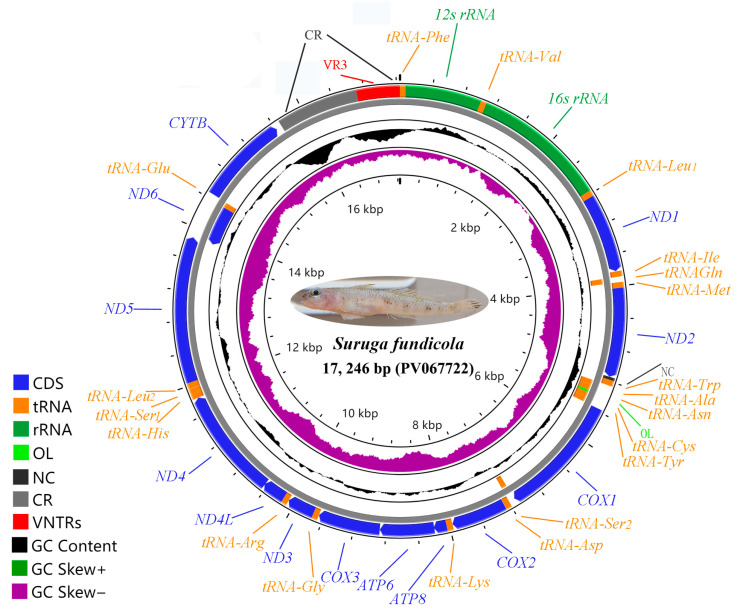
The circular mitogenome map of *S. fundicola* (PV067722). Genes encoded by the H-strand and L-strand are displayed in the outer and inner rings, respectively.

**Figure 3 ijms-26-08317-f003:**
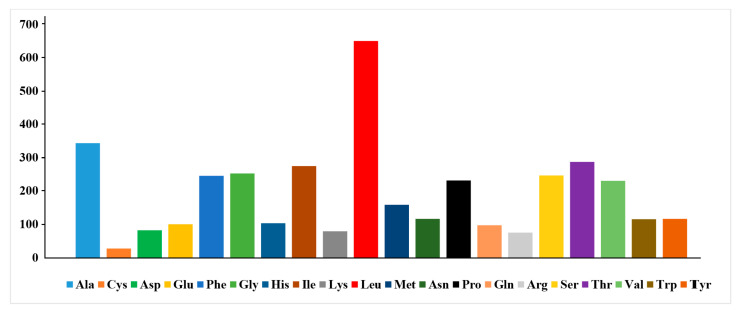
The amino acid distribution of the *S. fundicola* mitogenome. On the horizontal axis and vertical axis are the number and types of amino acids, respectively.

**Figure 4 ijms-26-08317-f004:**
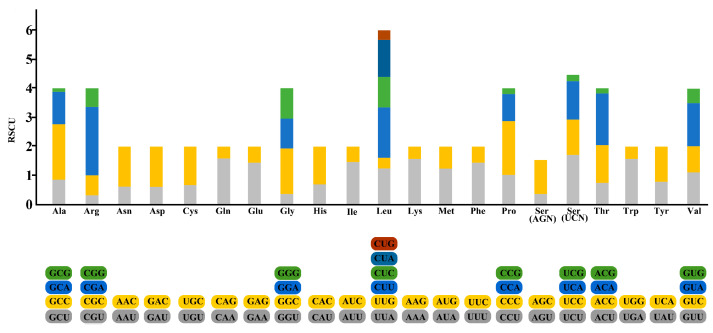
The relative synonymous codon usage (RSCU) of all 13 mitochondrial protein-coding genes of *S. fundicola*.

**Figure 5 ijms-26-08317-f005:**
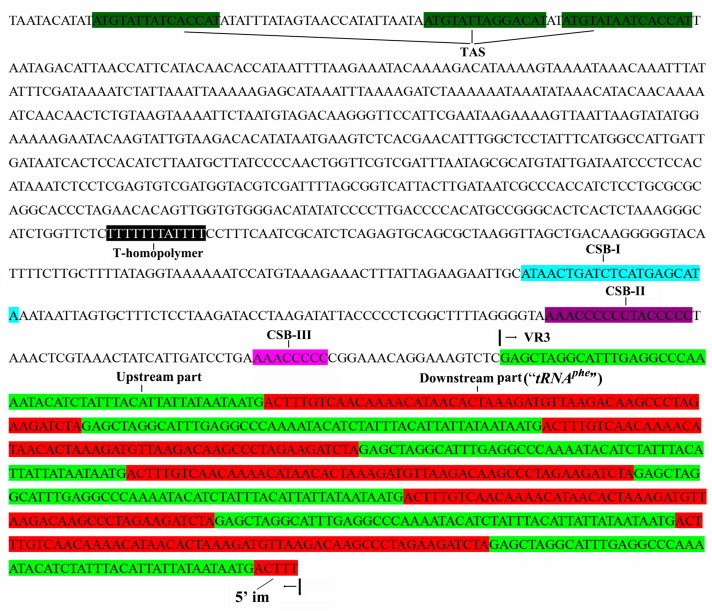
The sequence of the mitochondrial control region of *S. fundicola* PV067722. TAS, T-homopolymer, CSB elements, and VR3 consist of two regions: the upstream and downstream parts.

**Figure 6 ijms-26-08317-f006:**
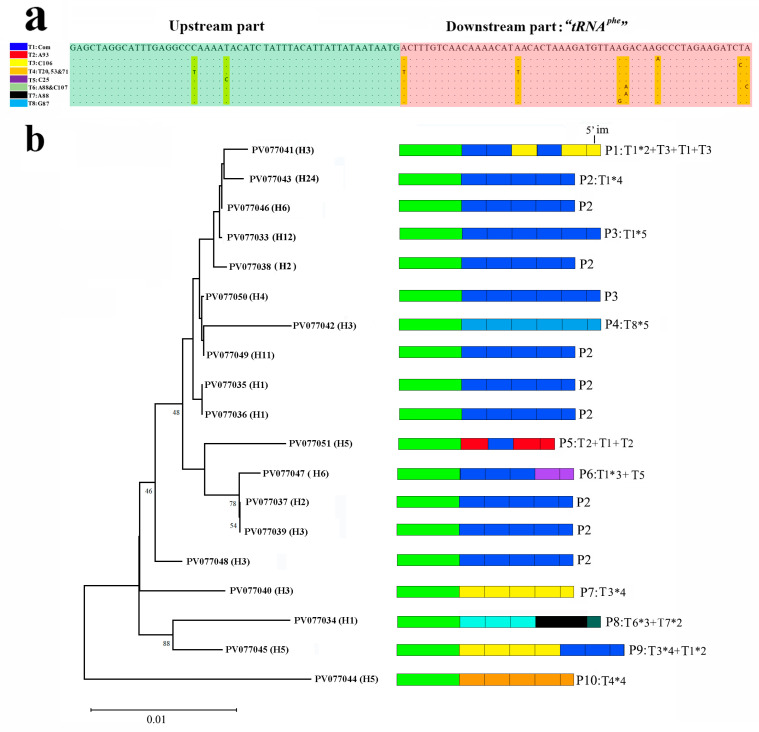
(**a**) The eight types of LTRs in the VR3 of *S. fundicola*; (**b**) the NJ tree constructed based on the VR3 of the LTR. The ten patterns of the LTRs found in the CR of the *S*. *fundicola* mitogenome are marked on the tree; the sampling location is listed in square brackets after the voucher number. The “*” represents multiplication and is used to indicate the repetition count of different types of LTRs.

**Figure 7 ijms-26-08317-f007:**
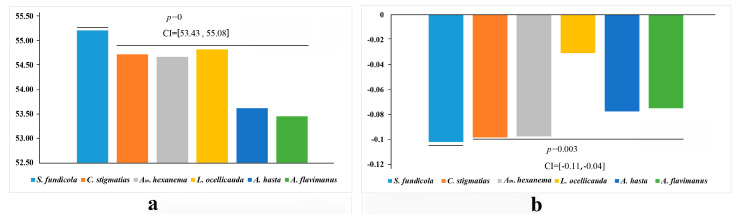
(**a**) The A + T content of species from the Acanthogobius Group; (**b**) the AT-skew of the same species. *p*-values and 95% confidence intervals (CIs) are noted.

**Figure 8 ijms-26-08317-f008:**
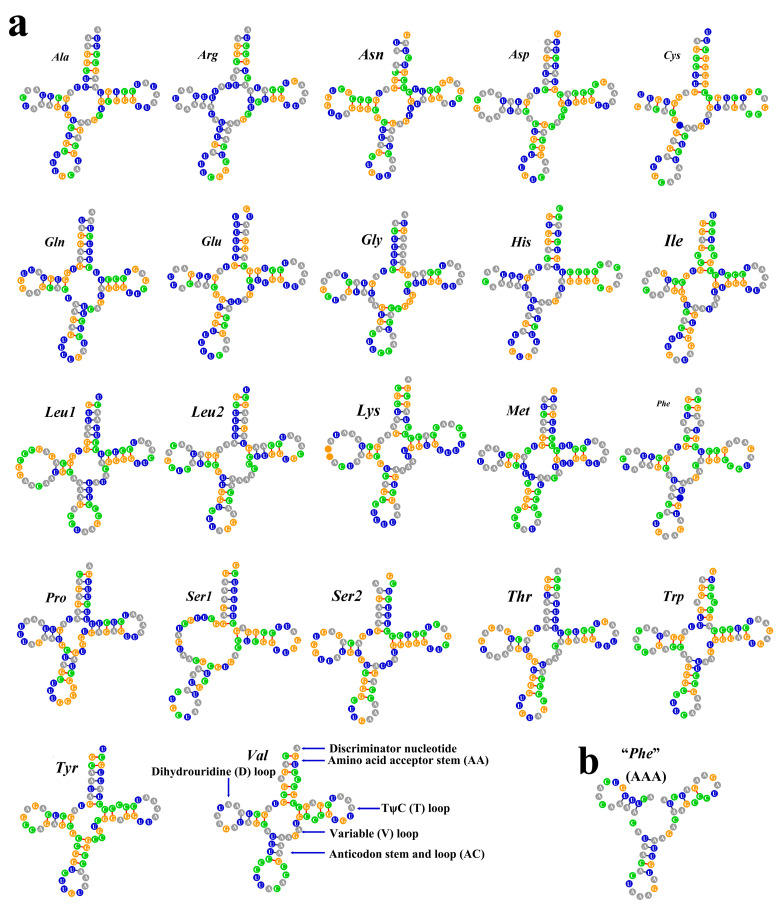
(**a**) The secondary structure of the 22 tRNA genes present in the mitogenome of *S. fundicola*; (**b**) the secondary structure of the putative tRNA*^Phe^* homologous sequence by tRNAscan-SE 2.0 in the CR.

**Figure 9 ijms-26-08317-f009:**
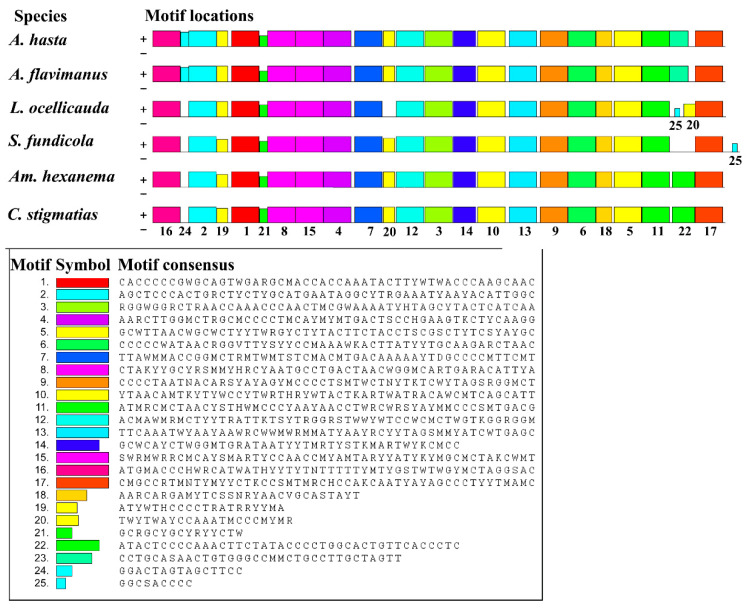
The motifs identified by MEME-SUITE in the *ND2* sequence of Acanthogobius Group mitogenomes.

**Figure 10 ijms-26-08317-f010:**
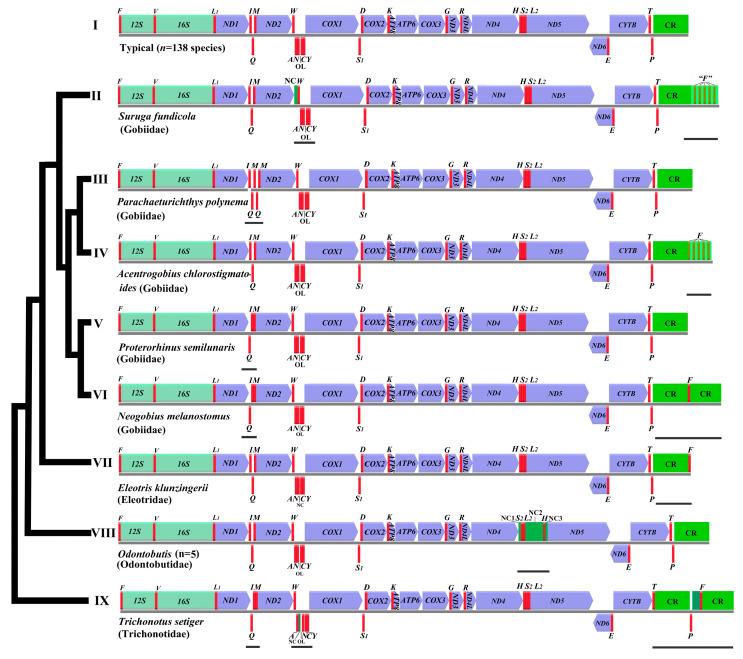
The linearized representation of the mitogenome of *S. fundicola* compared to other gobies. The mitochondrial genes, including 22~27 tRNAs (represented by a single uppercase italic letter), 2 mitochondrial rRNAs (12S and 16S), and 13 protein-coding genes, were labeled in red, light green, and purple, respectively. NC (noncoding sequence of ≥30 bp) was marked in green. “*F*”: the *tRNA^Phe^* homologous sequence identified by tRNAscan-SE 2.0. CR: control region. The variant regions are marked with black underlines.

**Figure 11 ijms-26-08317-f011:**
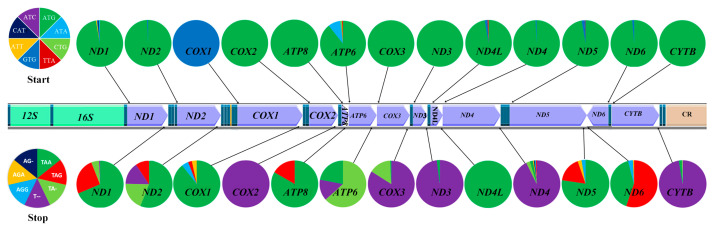
The usage bias of start and stop codons of 13 PCGs in 150 goby mitogenomes. Pie graphs show the frequency of the occurrence of the start and stop codons. Start codons (upper section) and stop codons (lower section) are represented by distinct colors, with the proportion of each color indicating the probability of occurrence of each respective codon across the 13 proteins.

**Figure 12 ijms-26-08317-f012:**
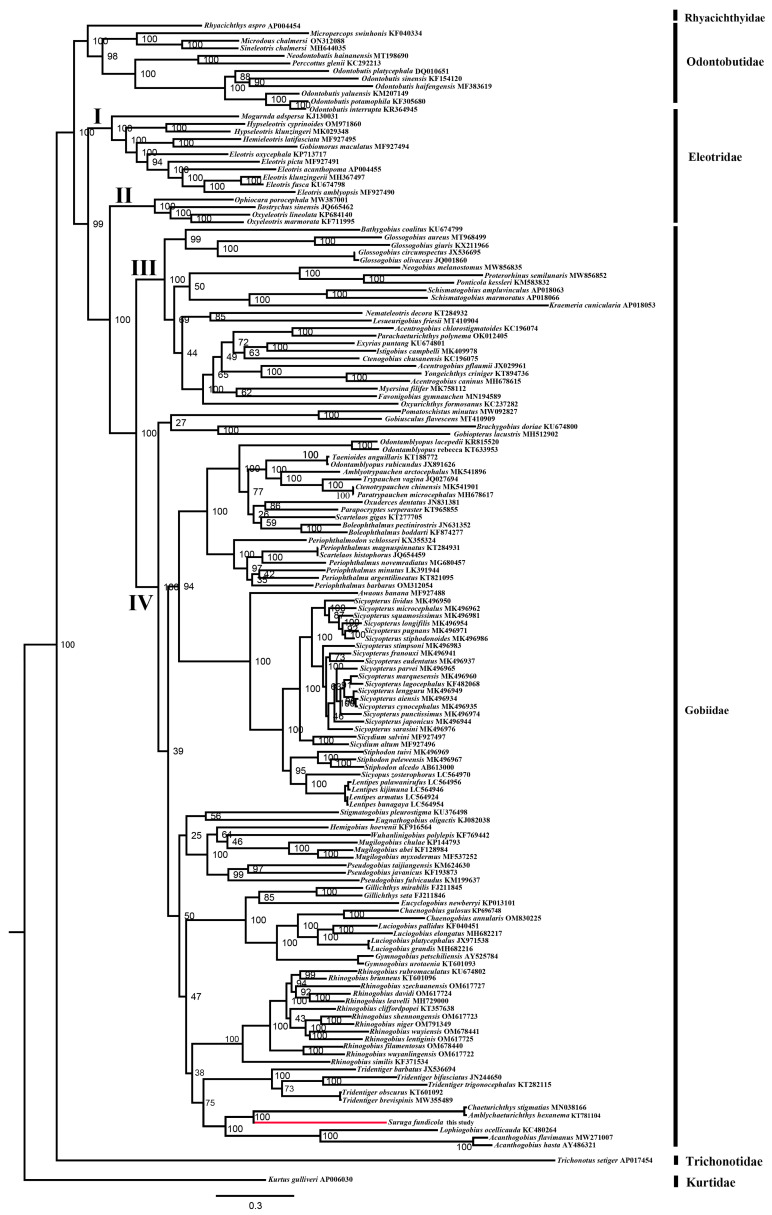
The ML tree was constructed based on concatenated sequences of 13 PCGs (without stop codon) of 151 mitogenomes. Bootstrap support values based on maximum likelihood analysis are shown (with 1000 replicates). The clade represented by *S. fundicola* is highlighted in red.

**Figure 13 ijms-26-08317-f013:**
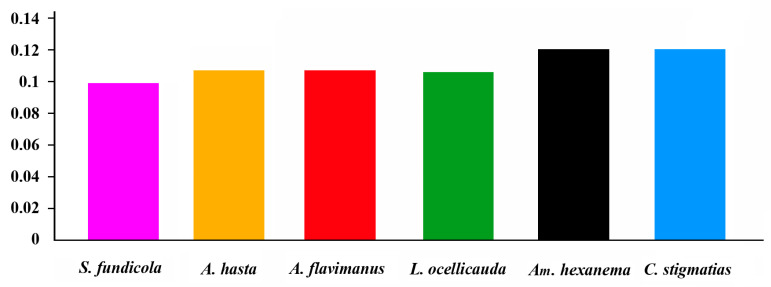
Individual Ka/Ks ratio values of the species of the Acanthogobius Group.

**Figure 14 ijms-26-08317-f014:**
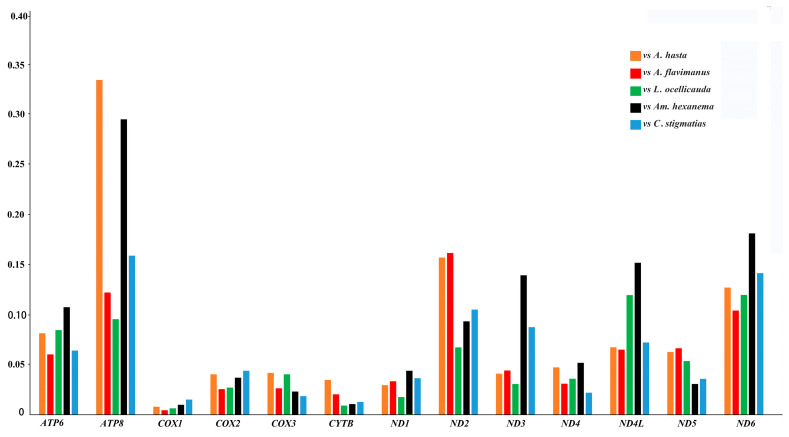
Analysis of selective pressure in the PCGs of *S. fundicola* vs. the other species of the Acanthogobius Group; the vertical axis shows the KA/KS ratio for each of the 13 protein-coding genes.

**Table 1 ijms-26-08317-t001:** The mitogenome features of *S. fundicola* (PV067722).

Gene	Strand	Location	Size (bp)	Anticodon	Start Codon	Stop Codon	Intergenics Length
*tRNA^Phe^ (S)*	+	1–68	68	GAA			0
*12srRNA*	+	69–1012	944				0
*tRNA^Val^ (V)*	+	1013–1083	71	TAC			0
*16srRNA*	+	1084–2761	1678				0
*tRNA^Leu(UUR)^ (L* *1* *)*	+	2762–2836	75	TAA			0
*ND1*	+	2837–3811	975		ATG	TAG	3
*tRNA^Ile^ (I)*	+	3815–3884	70	GAT			−1
*tRNA^Gln^ (Q)*	−	3884–3954	71	TTG			−1
*tRNA^Met^ * *(M)*	+	3954–4023	70	CAT			1
*ND2*	+	4025–5146	1122		ATG	TAA	0
NC	+	5147–5181	35				
*tRNA^Trp^ (W)*	+	5183–5253	71	TCA			2
*tRNA^Ala^ (A)*	−	5256–5324	69	TGC			1
*tRNA^Asn^ (N)*	−	5326–5398	73	GTT			33
OL	−	5399–5431					
*tRNA^Cys^ (C)*	−	5432–5496	65	GCA			0
*tRNA^Tyr^ (Y)*	−	5496–5566	71	GTA			1
*COX1*	+	5569–7122	1554		GTG	TAA	0
*tRNA^Ser^ (S* *2* *)*	−	7123–7193	71	TGA			3
*tRNA^Asp^ (D)*	+	7197–7268	72	GTC			3
*COX2*	+	7272–7962	691		ATG	T–	0
*tRNA^Lys^ (K)*	+	7963–8036	74	TTT			1
*ATP8*	+	8038–8202	165		ATG	TAA	−5
*ATP6*	+	8196–8878	683		ATG	TA–	0
*COX3*	+	8879–9662	784		ATG	T–	0
*tRNA^Gly^ (G)*	+	9663–9734	72	TCC			0
*ND3*	+	9735–10,083	349		ATG	T–	0
*tRNA^Arg^ (R)*	+	10,084–10,152	69	TCG			0
*ND4L*	+	10,153–10,449	297		ATG	TAA	−5
*ND4*	+	10,443–11,823	1381		ATG	T–	0
*tRNA^His^ (H)*	+	11,824–11,891	68	GTG			0
*tRNA^Serr(AGY)^ (S* *1* *)*	+	11,892–11,959	68	GCT			3
*tRNA^Leu(CUN)^ (L* *2* *)*	+	11,963–12,035	73	TAG			0
*ND5*	+	12,036–13,874	1839		ATG	TAA	−2
*ND6*	−	13,871–14,392	522		ATG	TAG	0
*tRNA^Glu^ (E)*	−	14,393–14,461	69	TTC			5
*CYTB*	+	14,467–15,610	1144		ATG	T–	0
*tRNA^Thr^ (T)*	+	15,611–15,682	72	TGT			−1
*tRNA^Pro^ (P)*	−	15,682–15,751	70	TGG			0
Control region	+	15,752–17,246	1495				–

**Table 2 ijms-26-08317-t002:** The nucleotide composition and strand asymmetry of the *S. fundicola* mitogenome.

	Length	T%	C%	A%	G%	AT%	AT- Skew%	GC- Skew%
PCGs	11,223	30.4	28.5	24.8	16.3	55.2	−0.10	−0.27
tRNA	1552	27.0	20.9	28.6	23.5	55.6	0.03	0.06
rRNA	2622	20.9	25.3	33.2	20.6	54.1	0.23	−0.10
CR	1495	28.8	19.3	37.7	14.2	66.5	0.13	−0.15
Genome	17,246	27.6	27.0	28.6	16.8	56.2	0.02	−0.23

**Table 3 ijms-26-08317-t003:** Positive selection in all 13 mitochondrial PCGs based on BM in EasyCodeML. The number with “*” indicates that the positive selection site has high confidence (*p* > 0.95).

	Model	Ln L	Model Compared	LRT *p*-Value	Positive Sites
*ATP6*	MA	−2587.829	MA vs. MA 0	0.266	7 D 0.960 *
	MA 0	−2588.449			Not Allowed
*ATP8*	MA	−726.169	MA vs. MA 0	1.000	
	MA 0	−726.169			Not Allowed
*COX1*	MA	−4856.932	MA vs. MA 0	1.000	
	MA 0	−4856.932			Not Allowed
*COX2*	MA	−2035.809	MA vs. MA 0	0.670	
	MA 0	−2035.718			Not Allowed
*COX3*	MA	−2355.767	MA vs. MA 0	1.000	
	MA 0	−2355.767			Not Allowed
*CYTB*	MA	−3942.845	MA vs. MA 0	1.000	
	MA 0	−3942.845			Not Allowed
*ND1*	MA	−3522.005	MA vs. MA 0	1.000	
	MA 0	−3522.005			Not Allowed
*ND2*	MA	−4337.265	MA vs. MA 0	0.385	2 N 0.987 *, 18 G 0.980 *, 21 A 0.980 *, 62 T 0.976 *, 240 T 0.967 *, 329 T 0.953 *, 345 L 0.982 *
	MA 0	−4337.643			Not Allowed
*ND3*	MA	−1202.254	MA vs. MA 0	0.346	88 T 0.952 *
	MA 0	−1202.698			Not Allowed
*ND4*	MA	−5157.592	MA vs. MA 0	1.000	351 S 0.957 *
	MA 0	−5157.592			Not Allowed
*ND4L*	MA	−1051.586	MA vs. MA 0	0.804	
	MA 0	−1051.617			Not Allowed
*ND5*	MA	−6756.556	MA vs. MA 0	0.472	
	MA 0	−6756.815			Not Allowed
*ND6*	MA	−2199.261	MA vs. MA 0	1.000	
	MA 0	−2199.261			Not Allowed

**Table 4 ijms-26-08317-t004:** PCR conditions for the two pairs of primers used in this study.

Region	Initial Denaturation	Denaturation	Annealing	Extension	Cycles	Volume
*ND2*	94 °C, 4 min	94 °C, 50 s	57 °C, 30 S	50 S	35	25 μL
VR3			50 °C, 30 S			

## Data Availability

The four complete mitogenomes of *S. fundicola* and four sequences of the *ND2* expansion were submitted to GenBank with the accession numbers PV067722-25 and PV067729-32. The nineteen VR3 sequences were submitted to GenBank with the accession number PV077033-51. Other supporting results are included within this article and its Appendix A.

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
