# Peer review of "The Complete Mitochondrial Genome of the Deep-Dwelling Goby Suruga fundicola (Teleostei, Gobiidae) Reveals Evidence of Recombination in the Control Region"

_ijms, 2025, doi:10.3390/ijms26178317_

Round 1
Reviewer 1 Report
Comments and Suggestions for Authors
This manuscript presented an investigation into the mitogenome of the deep-sea goby Suruga fundicola. Special features, were founded in the mitogenome, such as an extended ND2, an unexpected NC, and various length of control region (CR). This research provided another evidence for recombination in the mtDNA CR of vertebrate.
The strong purifying selection and potential positive selection sites identified in this species indicated adaptive evolution in mitochondrial PCGs in response to deepwater conditions. However, there are some weaknesses in this manuscript.
The authors observed potential recombination in the mitochondrial control region through length heterogeneity in long tandem repeats (LTR), suggesting that this may be attributed to paternal leakage during zygote formation, where mitochondria carrying two distinct LTR types coexist within a single cell. However, the hypothesis regarding recombination remains unexplored in depth.
The conclusion is too lengthy and complicated, and some conclusions were omitted, such as mitochondrial arrangement. Bedsides, several specific details during the manuscript writing process still require to be carefully considered, which are listed as follows:
- Give the whole words when PCGs appears in the first time in abstract and manuscript
- Line 61 add the citation number after Satoh (2016); Give the whole words for CSBs and VRs
- Line 81 change to “(depth about 40–400 m)”
- Maintain consistent capitalization, e.g., in terms like 'tRNAtrp' and 'TrnaTrp'
- Line 100 Result and Discussion
- Line 104 “accession numbers PV067722-PV067725”; same to Line 111, Line171
- Fig. 3 add axis labels to replace this sentence in the caption “The horizontal axis and vertical axis are the number and types of amino acids, respectively
- Line 171: sequenced 19 individuals “was” variable
- Fig. 6a: Inconsistent font sizes
- Fig. 6b: Missing in-text citation, and use accession numbers instead of specimen numbers;
- Line 208: "analysis" should be italicized
- Fig. 7: Add axis labels (x and y axes)
- Line 252: Inconsistent formatting of "OL" (vs. OL)
- Line 253: was not identified (Fig. 10 type VII)
- Figure 10: show what does the T(n=138) mean under type I
- Section 3.4: The analytical method is missing; please add it
- Fig. 10: Add in-text citation
- Fig. 13 and 14 add axis labels
- Line 268, CR: Define as "putative control region" consistently throughout
- Line 304: Unpaired parenthesis in "22-24"
- Table 4. Please give the PCR amplification volume and condition for these two primer pairs
- Line 415 Mega change to MEGA
- I did not see the Supplementary Materials in the review system, please make sure it will be uploaded in the later
Author Response
|
Response to Reviewer 1 Comments
|
||
|
1. Summary |
|
|
|
Thanks so much for your constructive advice and elaborated rigorous comments. In this study, we detected evidence suggestive of mitochondrial recombination within the Control Region (CR) through heterogeneity patterns in Long Terminal Repeats (LTRs). As you rightly noted, while our data indicate signs of recombination, the underlying mechanisms and processes require deeper investigation. We will evaluate advanced methodologies to probe recombination dynamics in future studies. Additionally, we refined the Conclusions section, and added the mitogenome rearrangement pattern of goby fish. See revised text in Lines 510–528.
|
||
|
2. Questions for General Evaluation |
Reviewer’s Evaluation |
Response and Revisions |
|
Does the introduction provide sufficient background and include all relevant references? |
Yes |
|
|
Are all the cited references relevant to the research? |
Yes |
|
|
Is the research design appropriate? |
Yes |
|
|
Are the methods adequately described? |
Yes |
|
|
Are the results clearly presented? |
Yes |
|
|
Are the conclusions supported by the results? |
Can be improved |
|
|
3. Point-by-point response to Comments and Suggestions for Authors
|
||
|
Comments 1: Give the whole words when PCGs appears in the first time in abstract and manuscript. |
||
|
Response 1: Thank you for pointing this out. We agree with this comment. Therefore, we have done as your mention, see line 35.
|
||
|
Comments 2: Line 61 add the citation number after Satoh (2016); Give the whole words for CSBs and VRs. |
||
|
Response 2: Agree. Thanks for your reminder, we did as your mention, see line 63.
Comments 3: Line 81 change to “(depth about 40–400 m)”. Response 3: Agree. We have changed the “meters” into “m”, see line 82.
Comments 4: Maintain consistent capitalization, e.g., in terms like 'tRNAtrp' and 'TrnaTrp'. Response 4: Agree. Response: thanks so much. We have changed “tRNAtrp” into “TrnaTrp”, and did the same for other tRNA genes, lines 223, 268, 493.
Comments 5: Line 100 Result and Discussion. Response 5: Agree. We have removed "+”, see line 101.
Comments 6: Line 104 “s PV067722-PV067725”; same to Line 111, Line171. Response 6: Agree. We have checked it, and revised, see lines 172, 186.
Comments 7: Fig. 3 add axis labels to replace this sentence in the caption “The horizontal axis and vertical axis are the number and types of amino acids, respectively. Response 7: Thanks, we have added the axis for the Figure 3.
Comments 8: Line 171: sequenced 19 individuals “was” variable. Response 8: Agree. Thanks, we have changed “is” into “was”.
Comments 9: Fig. 6a: Inconsistent font sizes Response 9: Agree. Thanks, we have adjusted the font sizes, see line 204 Figure 6.
Comments 10: Fig. 6b: Missing in-text citation, and use accession numbers instead of specimen numbers. Response 10: Agree. Thanks, we have added the in-text citation of Fig. 6b, and changed the GenBank accession numbers instead of specimen numbers.
Comments 11: "analysis" should be italicized. Response 11: Agree. Thanks, we have revised it, line 208.
Comments 12: Add axis labels (x and y axes) Response 12: Agree. Thanks, thanks, we have added the axis, see line 216 Fig. 7.
Comments 13: Inconsistent formatting of "OL" (vs. OL). Response 13: Agree. Thanks, we have chosen “OL”, see lines 152, 252.
Comments 14: was not identified (Fig. 10 type VII) Response 14: Agree. Thanks, we have added the word “type”.
Comments 15: show what does the T(n=138) mean under type I Response 15: Agree. Thanks, I am sorry for made this ambiguity. the T means typical linearized representation type of goby fishes, and n=138 means there were 138 species occupy this type. And we changed “T(n=138)” into “Typical (n=138 species)”.
Comments 16: The analytical method is missing; please add it Response 16: Agree. Thanks, we have found this, and have provided the first revised manuscript at July 19, 2025 throw Ms. Kayla Yu.
Comments 17: Add in-text citation Response 17: Agree. Thanks, we have added the in-text citation, see line 254.
Comments 18: Fig. 13 and 14 add axis labels Response 18: Agree. Thanks, we have added axis for Fig. 13 and 14, see line 254.
Comments 19: Line 268, CR: Define as "putative control region" consistently throughout Response 19: we have canceled “putative” for consistency.
Comments 20: Line 304: Unpaired parenthesis in "22-24" Response 20: Agree. Thanks, we add the “]” after 22-24, line 303.
Comments 21: Please give the PCR amplification volume and condition for these two primer pairs Response 21: Agree. Thanks, we have provided the PCR condition for the two pairs of primers, see Tab. 4 (line 388). Besides, we revised the section, see 368-379.
Comments 22: Line 415 Mega change to MEGA Response 22: Agree. Thanks, we have done as you mentioned.
Comments 23: I did not see the Supplementary Materials in the review system, please make sure it will be uploaded in the late Response 23: Agree. Thanks for your reminder, the attachment will be uploaded to review system.
|
||
Reviewer 2 Report
Comments and Suggestions for Authors The values presented between lines 114 and 120 are different from those presented in Table 1. In the text the sum is 99.9% and in the table = 100%.
The values presented between lines 114 and 120 are different from those presented in Table 1. In the text the sum is 99.9% and in the table = 100%. After this minimal correction I recommend the manuscript for publication.
Author Response
|
1. Summary |
|
|
|
Thank you for your positive comments and valuable suggestions, and we are so glad to gain your recognition. The specific modification details are as follows.
|
||
|
2. Questions for General Evaluation |
Reviewer’s Evaluation |
Response and Revisions |
|
Does the introduction provide sufficient background and include all relevant references? |
Yes |
|
|
Are all the cited references relevant to the research? |
Yes |
|
|
Is the research design appropriate? |
Yes |
|
|
Are the methods adequately described? |
Yes |
|
|
Are the results clearly presented? |
Yes |
|
|
Are the conclusions supported by the results? |
Yes |
|
|
3. Point-by-point response to Comments and Suggestions for Authors
|
||
|
Comments 1: The values presented between lines 114 and 120 are different from those presented in Table 1. In the text the sum is 99.9% and, in the table, = 100%. |
||
|
Response 1: Thanks for your reminder. My apologies for the error resulting from our negligence. This numbers have been revised as suggested, see lines 127-128, 134. |
||
Reviewer 3 Report
Comments and Suggestions for Authors
Review on “The complete mitochondrial genome of the deep-dwelling goby Suruga fundicola (Teleostei, Gobiidae) reveals evidence of recombination in the control region” for manuscript ID ijms-3784074
In this manuscript the authors describe firstly sequenced mt-genome of the goby Suruga fundicola, which improves understandings of phylogenetic relationship, mitogenome recombination, and adaptive evolution of goby. The intro section underscores the limited available mitochondrial genomic data for deep-water gobies, motivating the need for comprehensive genomic analysis to elucidate their evolutionary history and adaptive mechanisms. The introduction section is lacking current knowledge about mt-genomes of goby (closer species of Suruga fundicola). What was the purpose of sampling 19 individuals?
Major points:
L166: Figure 5 could be clearer: mtDNA Downstream parts aren’t contrast. The sequence ID in the upper part could be omitted.
L240-241: this sentence fits better to Methods.
L311-312: please specify the bootstrap value method
L414-436: why the phylogenetic tree was built by nucleotide sequences instead of protein sequences?
L354-355: the sampling details are required
L366-368: how primer design was performed? Please specify the software/method used for it.
L313-345: the role of mitochondrial genes in adaptation for deep-water and shallow water species could be explained in this section.
L500: supplementary files are missing in the submission
L526: These records has not yet been released in GenBank, please carefully check all the accession IDs.
Minor points:
L72: Ref. [25] covers the Chaeturichthys stigmatias and Amblychaeturichthys hexanema species only, collective name “Acanthobius” isn’t mentioned there.
L100: remove “+”
Fig 13: the color legend is useless here; the species caption could be made directly to each column.
Please use contrast colors in Fig 14 for A. hexanema and L. ocellicauda, the quality of picture would be better.
Reviewer 4 Report
Comments and Suggestions for Authors
Dear Authors,
The article “The complete mitochondrial genome of the deep-dwelling goby Suruga fundicola (Teleostei, Gobiidae) reveals evidence of recombination in the control region” introduces a genomic study of the deep-dwelling fish.
The are some omissions:
- Introduction section does not sufficiently disclose the problem, i.e. the reason to explore the complete mitochondrial genome of this rare goby.
- What is the practical application of the results obtained in this study? Why did you choose to study this particular organism ( fundicola)?
- When conducting the comparative analysis, it is essential to indicate p-value and level of confidence (for example, in figure 7).
Round 2
Reviewer 4 Report
Comments and Suggestions for Authors
Dear Authors,
The article has been improved.
It is desirable to include the highlights from Response 2 to the Introduction or Conclusion section
Author Response
|
1. Summary |
|
|
|
Once again, thank you for taking the time to continue providing valuable insights to enhance the quality of this manuscript. We have adopted your suggestion and incorporated the content of Response 2 into both the Introduction and Conclusion sections. This enhancement will further clarify the necessity and significance of our research. All revisions in the manuscript are highlighted in yellow.
|
||
|
2. Questions for General Evaluation |
Reviewer’s Evaluation |
Response and Revisions |
|
Does the introduction provide sufficient background and include all relevant references? |
Can be improved |
|
|
Are all the cited references relevant to the research? |
Yes |
|
|
Is the research design appropriate? |
Yes |
|
|
Are the methods adequately described? |
Yes |
|
|
Are the results clearly presented? |
Yes |
|
|
Are the conclusions supported by the results? |
Can be improved |
|
|
3. Point-by-point response to Comments and Suggestions for Authors
|
||
|
Comments 1: It is desirable to include the highlights from Response 2 to the Introduction or Conclusion section.
|
||
|
Response 1: Agree. Thanks so much. The content of Response 2 has been incorporated into Introduction (lines 68–71) and Conclusion (lines 532–534, lines 543–544, and lines 547–548). |
||